# Short Rotation Eucalypts: Opportunities for Biochar

**Donald L. Rockwood [1,2,\*], Martin F. Ellis [3], Ruliang Liu [4], Fengliang Zhao [4], Puhui Ji [4], Zhiqiang Zhu [4], Kyle W. Fabbro [2], Zhenli He [4]**  **and Ronald D. Cave [4]**

1   Florida FGT LLC, Gainesville, FL 32635, USA; floridafgt@cox.net
2   School of Forest Resources and Conservation, University of Florida, Gainesville, FL 32611, USA; kwfabbro@gmail.com
3   Green Carbon Solutions (GCS), Pepper Pike, OH 44124, USA; mellis@greencarbonsolutions.com
4   Indian River Research and Education Center (IRREC), University of Florida, Ft Pierce, FL 34945, USA; ruliang_liu@126.com (R.L.); zfl7409@163.com (F.Z.); jipuhui1983@163.com (P.J.); zhuzhiq8@163.com (Z.Z.); zhe@ufl.edu (Z.H.); rdcave@ufl.edu (R.D.C.)
*   Correspondence: dlrock@ufl.edu; Tel.: +01-352-256-3474

**Abstract:** Eucalypts can be very productive when intensively grown as short rotation woody crops (SRWC) for bioproducts. In Florida, USA, a fertilized, herbicided, and irrigated cultivar planted at 2471 trees/ha could produce over 58 green mt/ha/year in 3.7 years, and at 2071 trees/ha, its net present value (NPV) exceeded $750/ha at a 6% discount rate and stumpage price of $11.02/green mt. The same cultivar grown less intensively at three planting densities had the highest stand basal area at the highest density through 41 months, although individual tree diameter at breast height (DBH) was the smallest. In combination with an organic fertilizer, biochar improved soil properties, tree leaf nutrients, and tree growth within 11 months of application. Biochar produced from *Eucalyptus* and other species is a useful soil amendment that, especially in combination with an organic fertilizer, could improve soil physical and chemical properties and increase nutrient availability to enhance *Eucalyptus* tree nutrition and growth on sandy soils. Eucalypts produce numerous naturally occurring bioproducts and are suitable feedstocks for many other biochemically or thermochemically derived bioproducts that could enhance the value of SRWCs.

**Keywords:** *Eucalyptus*; short rotation woody crops; Florida; biochar; bioproducts

## 1. Introduction

Eucalypts are the world's most valuable and widely planted hardwoods (up to 21.7 million ha in 61 countries by 2030 [1]) and have numerous potential applications as short rotation woody crops (SRWCs) [2,3]. Several *Eucalyptus* planting stocks have promise as SRWCs in Florida [4,5], including *E. grandis* x *E. urophylla* cultivars such as EH1. After four generations of *E. grandis* genetic improvement for Florida's unique climatic and edaphic conditions starting in the 1960s and clonal testing initiated in the 1980s across a wide range of site/soil types, the University of Florida released five *G Series* cultivars in 2009 for commercial planting [4,5]. Although G1 is no longer commercially viable due to susceptibility to blue gum chalcid (*Leptocybe invasa*), G2 through G5 have shown resilience to damaging freezes, tolerance to infertile soils, exceptional stem form, improved coppicing ability, chalcid resistance, and varying degrees of windfirmness.

*G Series* planting density trials established on former citrus lands and phosphate mined clay settling areas in central and south Florida demonstrated maximum mean annual increments (MAI_max) as high as 75.3 to 78.2 green mt/ha/year at 4304 and 5066 trees/ha, respectively [6]. Economic analyses using current stumpage prices, high silvicultural management costs, and expected coppice yields, have shown that *G Series* cultivars can generate internal rates of return greater than 10% [6].

Most soils in Florida are sandy, with >90% of soil particles as sand, and have low nutrient and moisture holding capacities. Fertilization is necessary to sustain desired crop yield and quality. However, fertilizers are readily leached if not taken up by crop plants and consequently result in environmental pollution such as eutrophication. Applying biochar, a fine-grained, highly porous "charcoal" produced through pyrolysis (burning in a nearly oxygen-free environment) or gasification of numerous feedstocks, improves the physicochemical properties of soils, including bulk density, porosity, cation exchange capacity (CEC), and pH. It also increases soil water and nutrient holding capacities and consequently influences crop production while reducing leaching [7]. Productivity of many crops significantly increased after soils were amended with biochar [8,9]. Sandy soils are more responsive to biochar than clayey soils [10] due to their low water and nutrient holding capacities [11].

Biochar, an ancient soil-building amendment, today has wide ranging applications [12]. The International Biochar Institute (www.biochar-international.org) has identified more than 50 uses for biochar, and worldwide interest in and demand for biochar are growing quickly. Current demand estimates suggest that biochar is a billion dollar plus industry worldwide with the two largest markets being North America and Europe [13]. Depending on its particular properties, effective biochar uses include soil and crop improvement plus environmental benefits such as carbon sequestration, retention of nutrients and water, reduced leaching, and water purification, all of which are important in Florida.

Using experience in Florida, USA, we describe eucalypts' potential for maximizing SRWC productivity through site amendment and genetic improvement, document their suitability for biochar production, and assess biochar's potential for improving soil properties, tree nutrition, and eucalypts' growth.

## 2. Materials and Methods

### 2.1. EH1 Planting Density Demonstration

An 8.1 ha, intensively fertilized, herbicided, and irrigated *E. grandis* x *E. urophylla* cultivar EH1 planting density demonstration was established in May 2011 near Hobe Sound, FL, as five rows of trees on 19.8m-wide former citrus beds. Planting densities of 2071 trees/ha (3.1 × 1.2 m spacing) and 1181 trees/ha (3.1 × 2.1 m) were monitored in 21 permanent 19.8 × 12.2 m plots through harvest in December 2017. To model EH1 yield at these two densities and estimate productivity at 2471 trees/ha, 18-, 30-, 36-, 42-, 48-, 65-, and/or 81-month data were fit to stand density, dominant height and basal area development functions in *E. grandis* stand-level growth, and yield model equations used by Plessis and Kotze [14]. Yields were the same for high and low management strategies because *E. grandis* productivity under low management on citrus beds in central Florida generated similar yields to the EH1 under high management at Hobe Sound, FL [6]. Silvicultural and other forest management costs were provided by agricultural companies exploring *Eucalyptus* options in central and southern Florida. Stumpage prices were based on local *Eucalyptus* mulchwood stumpage prices reported by the same companies and a forestry consulting firm familiar with the local markets.

Maximum net present values ($NPV_{max}$) calculated for two management strategies (Table 1), two planting densities (2071 and 1181 trees/ha), two real discount rates (6% and 8%), and two stumpage prices ($11.02 and 22.05/green mt) assumed three stages (two coppice stages following the original planting) in one planting cycle. Based on young coppice in Evans Properties' EH1 plantation near Ft. Pierce, FL, expected coppice yields for stages 2 and 3 (first and second coppice, respectively) were assumed to be 90% and 80% of observed stage 1 yields, respectively. The optimum stage lengths were reported to the nearest 1/10th year; therefore, the annual interest rate was converted to an effective periodic rate.

**Table 1.** Low and high management activities and assumed costs for *E. grandis* x *E. urophylla* cultivar EH1 on former bedded citrus sites.

| Activity | Cost |
| --- | --- |
| Land Preparation | $988/ha |
| Chemical Site Prep | $297/ha |
| Propagules | $0.70/tree |
| Planting Cost | $0.40/tree |
| Irrigation + Fertilization (High management only) | $1977/ha |
| Fertilization at Initial Establishment (Low management only) | $173/ha |
| Weed Control (Beginning of coppice stage) | $136/ha |

*2.2. EH1 Fertilizer x Planting Density Study*

Cultivar EH1 was also planted in a fertilizer x planting density study in June 2015 on a former pasture at the Indian River Research and Education Center (IRREC) near Ft Pierce, FL. Five fertilizers (control, Green Edge (GE) 6–4–0 + micronutrients at 112, 224, and 336 kg of N/ha rates, and diammonium phosphate equivalent to 336 kg of N/ha) were applied as five treatment plots of three contiguous rows 3.1 m apart, for a total of 15 rows of 26 trees. Within the 26-tree rows, 5-tree row plots were systematically assigned one of three planting densities (3588, 1794, and 1196 trees/ha; 3.1 × 0.9 m, 1.8 m, and 2.7 m, respectively) such that 1794 and 1196 trees/ha were replicated twice, 3588 trees/ha once. The interior three trees of each fertilizer x planting density plot were periodically measured through November 2018. Analyses of variance and Duncan's Multiple Range Tests of fertilizer and planting density means were conducted using SAS® (SAS Institute, Cary, NC, USA).

*2.3. Biochar Tests*

Five test trees were used for preliminary biochar evaluations in 2017–2018. One tree in the EH1 Planting Density Demonstration, one *E. grandis* cultivar G2 in a 2012 commercial plantation near Ft Pierce, FL, one *Corymbia torelliana* tree in an adjacent progeny test, one *E. amplifolia* in a progeny test near Old Town, FL, and one *Quercus virginiana* in a nearby natural forest each provided ~23 green kg of stemwood for testing by a lab in California at different pyrolysis temperatures to determine optimal charring temperatures for the different feedstocks. Their biochar physical and chemical properties were further tested by Celignis Analytical, Ireland, to guide the processing of biochar and as a comparative benchmark.

Biochar produced in Europe by Green Carbon Solutions' (GCS') Polchar, which specializes in pyrolysis and carbonization of different feedstocks, served as a comparison for the five Florida trees. Hardwood monoculture roundwood logs only were cut to size, split, and pre-dried. Pyrolysis involved a vertical retort operating through a range of temperatures up to a maximum of ~630 °C. Post production, the biochar was sampled and tested for physical and chemical properties. Polchar's biochar was also used for the biochar–fertilizer study described below.

*2.4. Biochar–Fertilizer Study*

A two-row windbreak study of three *E. grandis* cultivars in one row and four *C. torelliana* progenies in an adjacent row offset 1.2 m away was established at the IRREC in July 2017 as a randomized complete block design with four complete replications of cultivars G3, G4, and G5 in 17 to 28-tree single row plots at 1.8 m within row spacing and one incomplete replication of cultivar G5 in a 13-tree single row plot. In February 2018, all four complete replications received an organic fertilizer (GE 6–4–0 + micronutrients at 336 kg of N/ha), and the two interior replications also received GCS' Polchar biochar (11.2 mt/ha) by rotovating the two treatments into the soil to a 20 cm depth between and within 1.2 m of the two rows. The incomplete replication served as a control. The cultivars in this resulting biochar–fertilizer study were measured at 5, 11, and 16 months.

To monitor soil and foliage responses, 13 trees (one in the middle of each of the 13 cultivar plots) were resampled at biochar–fertilizer treatment ages of 0, 5, and 11 months (tree ages 5, 11, and 16 months, respectively). At each time, four soil samples were collected from a 0–20 cm depth within 1.2 m around each sample tree, and recently matured foliage was taken from four representative branches around the crown of each sample tree. The collected soils were combined by tree, air dried, and ground to pass through a 2-mm sieve prior to analysis for relevant properties. The tree leaf samples were combined by tree, oven dried at 75 °C to constant weight, and powdered to pass through a 1-mm sieve prior to analysis for nutrient concentration.

Soil pH was measured using a pH/conductivity meter (AB 200, Fisher Scientific, Pittsburgh, PA, USA) at the soil to water ratio of 1:1. Electrical conductivity (EC) of soil samples was determined at the solid to water ratio of 1:2 using the pH/conductivity meter. Available soil P was determined using the method of Kuo [15]. Available metals in soil were measured by extracting the samples with Mehlich 3 (M3) solution at a solid to solution ratio of 1:10 [16]. The extracts were filtered through a 0.45-μm membrane. Subsamples of the filtrate were acidified and analyzed for the concentrations of dissolved P, K, Ca, Mg, Fe, Mn, Cu, Zn, B, and Mo using inductively coupled plasma–optical emission spectrometry (ICP–OES) (Ultima, JY Horiba Group, Edison, NJ). Portions of the plant leaf samples (0.2 g each) were digested with 6 mL of concentrated nitric acid ($HNO_3$)/hydrogen peroxide ($H_2O_2$) and diluted to 100 mL. The concentrations of P, K, Ca, Mg, Fe, Mn, Cu, Zn, B, and Mo in the digested samples were determined using ICP–OES.

Analyses of variance and Tukey–Kramer tests of cultivar tree size, soil nutrient, and tree leaf nutrient means were conducted using SAS®. Changes between soil properties and leaf nutrients from 0–5, 5–11, and 0–11 months were also analyzed.

## 3. Results

### 3.1. EH1 Planting Density Demonstration

Through 81 months, cultivar EH1 yielded more at 2071 trees/ha than at 1181 trees/ha. Maximum annual yields were directly related, and times to those peaks were inversely related, to planting density (Table 2). Annual yield at 2471 trees/ha was estimated to be over 58 green mt/ha/year in 3.7 years. At the lowest density, $MAI_{max}$ was 44 mt/ha/year at 5.0 years. Compared to 2471 trees/ha, 2071 trees/ha had lower yields and achieved peak productivity later. However, planting density also inversely affected average tree diameter at breast height (DBH) as the higher planting density produced smaller trees (e.g., 81-month DBH = 15.4 cm at 2071 trees/ha, 19.2 cm at 1181 trees/ha), which could influence harvesting costs.

**Table 2.** Predicted maximum mean annual increment ($MAI_{max}$) and rotation age for cultivar EH1 at planting densities of 1181, 2071, and 2471 trees/ha.

| Planting Density (trees/ha) | $MAI_{max}$ (green mt/ha/year) | Rotation Age at $MAI_{max}$ (years) |
| --- | --- | --- |
| 1181 | 44.00 | 5.0 |
| 2071 | 54.63 | 4.0 |
| 2471 | 58.98 | 3.7 |

NPVs ranged widely largely due to stumpage price (Table 3). At a stumpage price of $11.02/mt, high management intensity had negative NPVs. At $22.05/mt stumpage, all scenarios resulted in positive NPVs. Due to high establishment and planting costs, 2071 trees/ha generated higher NPVs compared to 1181 trees/ha across all scenarios. Stage lengths were always shorter with 2071 trees/ha and increasing discount rate, which for example could result in a ~15.5 year planting cycle consisting of 4.8 years from planting to first harvest, 5.1 year first coppice stage, and 5.6 year second coppice stage.

**Table 3.** Effects of stumpage price ($/green mt) and real discount rate on maximum net present values (NPV$_{max}$), internal rate of return (IRR), and optimum stage lengths for the first cycle of EH1 at 2071 and 1181 trees/ha with low and high management intensities and two coppice stages.

| Stumpage Price ($) | Discount Rate (%) | NPV$_{max}$ ($/ha) | IRR (%) | Stage Length (years) |
|---|---|---|---|---|
| **2071 trees/ha, Low Management Intensity ($1458/ha + $1.10/tree + $136/ha @coppice)** | | | | |
| 11.02 | 6 | 751 | 8.1 | 5.1, 5.4, 5.8 |
| | 8 | 70 | 8.2 | 4.9, 5.1, 5.6 |
| 22.05 | 6 | 5365 | 18.1 | 5.1, 5.4, 5.8 |
| | 8 | 3986 | 18.6 | 4.8, 5.1, 5.6 |
| **2071 trees/ha, High Management Intensity ($3262/ha + $1.10/tree + $136/ha @coppice)** | | | | |
| 11.02 | 6 | −1053 | 3.6 | 5.1, 5.4, 5.8 |
| | 8 | −1734 | 3.5 | 4.9, 5.1, 5.6 |
| 22.05 | 6 | 3561 | 12.2 | 5.1, 5.4, 5.8 |
| | 8 | 2182 | 12.5 | 4.8, 5.1, 5.6 |
| **1181 trees/ha, Low Management Intensity ($,458/ha + $1.10/tree + $136/ha @coppice)** | | | | |
| 11.02 | 6 | 1377 | 10.3 | 6.3, 6.7, 7.4 |
| | 8 | 660 | 10.5 | 5.9, 6.3, 7.0 |
| 22.05 | 6 | 5509 | 19.2 | 6.3, 6.7, 7.4 |
| | 8 | 4058 | 19.9 | 5.9, 6.2, 7.0 |
| **1181 trees/ha, High Management Intensity ($3262/ha + $1.10/tree + $136/ha @coppice)** | | | | |
| 11.02 | 6 | −427 | 5.0 | 6.3, 6.7, 7.4 |
| | 8 | −1143 | 4.9 | 5.9, 6.3, 7.0 |
| 22.05 | 6 | 3705 | 12.4 | 6.3, 6.7, 7.4 |
| | 8 | 2254 | 12.7 | 5.9, 6.2, 7.0 |

## *3.2. EH1 Fertilizer x Planting Density Study*

Fertilizer and planting density influenced the productivity of cultivar EH1 (Table 4). While the differences among five fertilizers for 9-mo height favored the higher GE rates, subsequent differences were inconsistent due to flooding soon after planting and small plot sizes. Planting density differences were observed at all ages, with 3588 trees/ha having the tallest trees at 9 months and the largest stand basal area but smallest tree DBH at subsequent ages. For example, at 41 months, stand basal area and tree DBH at 3588 trees/ha averaged 31.9 m$^2$/ha and 10.3 cm, respectively, compared to 19.6 m$^2$/ha and 14.2 cm at 1196 trees/ha.

**Table 4.** Effects of three planting densities (trees/ha) and five fertilizers on EH1 tree height (m), diameter at breast height (DBH) (cm), and stand basal area (m$^2$/ha) at 9, 36, and 41 months at the Indian River Research and Education Center (IRREC).

| Trait: Age | Planting Density | Fertilizer | | | | | Density Average |
|---|---|---|---|---|---|---|---|
| | | 0 | GE 100 | GE 200 | GE 300 | DAP | |
| Height 9-mo | 3588 | 3.86 | 4.62 | 5.66 | 5.02 | 3.36 | 4.60a |
| | 1794 | 2.83 | 2.80 | 3.76 | 3.54 | 2.97 | 3.17ab |
| | 1196 | 3.81 | 3.35 | 5.03 | 4.41 | 3.65 | 4.02ab |
| | Fert. Ave. | 3.43b | 3.43b | 4.71a | 4.24ab | 3.33b | 3.83 |
| DBH 36-mo | 3588 | 9.4 | 9.8 | 9.8 | 10.8 | 6.7 | 9.3b |
| | 1794 | 10.6 | 9.6 | 11.8 | 10.8 | 12.0 | 10.9ab |
| | 1196 | 14.8 | 12.2 | 14.7 | 12.1 | 12.6 | 13.2ab |
| | Fert. Ave. | 11.8 | 10.7 | 12.6 | 11.3 | 11.2 | 11.5 |
| Basal Area 36-mo | 3588 | 27.3 | 27.1 | 31.5 | 33.0 | 13.0 | 26.5a |
| | 1794 | 16.1 | 15.0 | 19.9 | 18.1 | 20.7 | 17.8b |
| | 1196 | 20.8 | 14.1 | 20.5 | 14.1 | 14.9 | 16.8b |
| | Fert. Ave. | 20.4 | 17.1 | 22.5 | 19.5 | 16.5 | 19.2 |
| DBH 41-mo | 3588 | 9.9 | 11.1 | 10.8 | 11.7 | 7.7 | 10.3b |
| | 1794 | 13.2 | 10.6 | 13.1 | 12.1 | 12.2 | 12.3ab |
| | 1196 | 16.0 | 11.9 | 15.3 | 13.6 | 14.1 | 14.2a |
| | Fert. Ave. | 13.7a | 11.2b | 13.2ab | 12.6ab | 12.0ab | 12.5 |
| Basal Area 41-mo | 3588 | 24.7 | 35.5 | 39.9 | 38.9 | 17.6 | 31.9a |
| | 1794 | 25.6 | 18.0 | 24.7 | 22.4 | 21.8 | 22.5b |
| | 1196 | 24.5 | 14.5 | 22.6 | 17.9 | 18.9 | 19.6c |
| | Fert. Ave. | 25.0a | 20.8b | 29.3a | 25.4a | 19.6b | 23.8 |

* Means within Fert. or Density Averages not sharing the same letter differ at the 5% level.

### 3.3. Biochar Tests

Biochars from EH1, *C. torelliana*, G2, *E. amplifolia*, and *Quercus virginiana*, were relatively similar and appeared suitable for commercial biochar production (Table 5). The Cl content of G2, though, was somewhat high. Compared to Polchar biochar made from European hardwoods, which was high quality with a pH of 8.2 and electrical conductivity of 3.33 mmhos/cm, all five Florida trees appeared similar for recalcitrant carbon but higher in pH and water holding (Table 6).

**Table 5.** Properties of Green Carbon Solutions (GCS) biochar made from Florida *E. grandis* cultivar G2, *C. torelliana* (CT), *E. grandis* x *E. urophylla* cultivar EH1, *E. amplifolia* (EA), and *Q. virginiana* (Qv) test trees.

| Property (% of Dry Weight) | G2 | CT | EH1 | EA | Qv |
|---|---|---|---|---|---|
| Volatile Matter | 83.3 | 85.0 | 85.9 | 82.5 | 83.3 |
| Fixed Carbon | 15.7 | 14.4 | 13.7 | 17.0 | 15.5 |
| Ash | 1.00 | 0.54 | 0.37 | 0.50 | 1.15 |
| Moisture Content | 36.4 | 48.0 | 43.1 | 30.1 | 33.1 |
| C | 49.2 | 49.7 | 49.8 | 50.8 | 49.1 |
| O | 43.0 | 43.1 | 43.1 | 42.0 | 43.1 |
| H | 6.5 | 6.5 | 6.5 | 6.5 | 6.4 |
| N | 0.21 | 0.17 | 0.17 | 0.26 | 0.29 |
| Cl | 0.07 | 0.02 | 0.02 | 0.02 | 0.00 |
| S | 0.01 | 0.00 | 0.00 | 0.01 | 0.00 |

**Table 6.** Comparison of properties of biochar made from Florida *E. grandis* cultivar G2, *C. torelliana* (CT), *E. urograndis* cultivar EH1, *E. amplifolia* (EA), and *Q. virginiana* (QV) test trees with Polchar biochar.

| Property | Florida Tree | | | | | Polchar Biochar |
|---|---|---|---|---|---|---|
| | G2 | CT | EH1 | EA | QV | |
| Recalcitrant Carbon * (%) | 76.0 | 71.6 | 74.0 | 70.8 | 71.8 | 67.6 |
| pH | 10.6 | 10.4 | 10.5 | 11.1 | 11.9 | 8.2 |
| EC (mmhos/cm) | 0.57 | 1.76 | 1.56 | 3.88 | 1.14 | 3.33 |
| Water Holding (mL/100 g) | 75.9 | 78.8 | 79.8 | 69.0 | 68.5 | 43.4 |
| Carbonate Value (%) | 2.6 | 2.5 | 5.6 | 16.7 | 2.5 | - |

* Estimated at 80% of fixed carbon on a dry ash-free basis.

### 3.4. Biochar–Fertilizer Study

Soil property data from the IRREC biochar–fertilizer study suggest that biochar (BC) enhanced the nutrient properties of this inherently poor Florida soil (Table 7). GE generally increased available soil nitrogen, as indicated by increases in KCl extractable $NO_3$-N and $NH_4$-N, and GE in combination with BC further increased $NH_4$-N five months after GE + BC application, which may be attributed to increased $NH_4$-N holding capacity in the GE + BC amended soil. An increase in soil available P was significant ($p = 0.0198$) for GE + BC five months after amendment. However, both available N and P in the soil decreased 11 months after amendment, likely due to intensive uptake by the established trees. At 11 months, soil $NH_4$-N was significantly ($p = 0.0376$) higher with GE and GE + BC compared to the control. Replication and cultivar effects were non-significant except for beginning EC and EC from beginning to 11 months (replications, $p = 0.0147$ and 0.0368, respectively), and 11-month EC and EC from beginning to 5 months (cultivars, $p = 0.0396$ and 0.0043, respectively).

**Table 7.** Effects of three cultural treatments (Green Edge only (GE), Green Edge with biochar (GE + BC), and Control) on soil properties before and after treatment applications in the IRREC biochar–fertilizer study.

| Treatment | Soil Property * | | | | |
|---|---|---|---|---|---|
| | pH | EC (uS/cm) | $NO_3$-N (mg/kg) | $NH_4$-N (mg/kg) | P (mg/kg) |
| **Before GE and GE + BC Applications** | | | | | |
| GE | 6.30 ± 0.73 | 46.1b ± 28.8 | 2.36 ± 0.51 | 0.80 ± 0.41 | 7.27 ± 3.24 |
| GE + BC | 5.32 ± 0.28 | 58.6a ± 23.1 | 4.08 ± 1.34 | 1.63 ± 0.74 | 7.26 ± 5.14 |
| **5 Months After GE and GE + BC Applications** | | | | | |
| GE | 6.25 ± 0.83 | 82.6 ± 28.9 | 3.51 ± 1.20 | 1.88 ± 1.07 | 7.84b ± 1.33 |
| GE + BC | 6.15 ± 0.69 | 96.1 ± 30.3 | 3.84 ± 1.31 | 2.17 ± 1.19 | 10.84a ± 2.42 |
| **11 Months After GE and GE + BC Applications** | | | | | |
| GE | 6.01 ± 0.55 | 27.1 ± 3.1 | 1.42 ± 0.27 | 0.85a ± 0.18 | 6.01 ± 1.09 |
| GE + BC | 5.96 ± 0.30 | 29.4 ± 2.5 | 2.15 ± 0.39 | 1.33a ± 0.30 | 6.26 ± 1.36 |
| Control | 5.44 ± --- | 21.3 ± --- | 0.83 ± --- | 0.32b ± --- | 2.01 ± --- |

* Mean ± Standard Deviation; *n* = 6 for GE and GE + BC, *n* = 1 for Control; Means within a Soil Property and time not sharing the same letter are different at the 5% level.

The IRREC biochar–fertilizer study leaf nutrient data suggest that biochar also enhanced the nutrient levels of the three *E. grandis* cultivars in the study (Table 8). Application of GE generally increased the concentrations of Ca, K, Mg, P, Fe, and Mn in tree leaves, especially for K, Mg, P, Fe, and Mn, which increased 1–4 times in five months after amendment; GE + BC significantly increased ($p = 0.0161$) 5-month Zn over GE. These increases are likely due to inputs of these nutrients in GE, thus improving their availability in soil. However, Zn and Cu concentrations decreased, which may be

attributed to binding of these elements to the organic components in GE, thus reducing their availability in the amended soil. Addition of BC to GE further improved tree nutrition with Ca, Mg, Zn, and Mn, and such improvement was also observed in 11 months after amendment, as BC tended to minimize leaf nutrient changes over time. However, a general decrease in leaf concentrations of Ca, K, Mg, P, and Zn also occurred, likely due to decreased availability of these nutrients in soil (Table 7) and the dilution effect of rapidly increased tree biomass. Replication and cultivar differences were detected only for initial P and Zn at 5 months (replications, $p = 0.0236$ and $0.0111$, respectively) and the change in Mg from 0 to 11 months (cultivars, $p = 0.0369$).

In the IRREC biochar–fertilizer study, GE and GE + BC gradually enhanced the growth of three *E. grandis* cultivars (Table 9). Before treatment applications at age 5 months, the cultivars were 0.8 to 1.3 m tall in the plots that then received three treatments. Six months after application, the cultivars receiving GE + BC and GE only had doubled in height, approximately twice the increase with no treatment. Eleven months after application, cultivars receiving GE + BC were 1.0 and 2.8 m taller than those receiving GE only and no GE. Based on the cultivar G5 common to all three cultures, tree height and DBH were then significantly greater with GE and GE + BC.

**Table 8.** Effects of three cultural treatments (Green Edge only (GE), Green Edge with biochar (GE + BC), and Control) on *E. grandis* cultivars leaf nutrients (Ca, K, Mg, and P in g/kg; Zn, Cu, Fe, and Mn in mg/kg) before and after treatment applications in the IRREC biochar–fertilizer study.

| Treatment | Leaf Nutrient * | | | | | | | |
|---|---|---|---|---|---|---|---|---|
| | Ca | K | Mg | P | Zn | Cu | Fe | Mn |
| *Before GE and GE + BC Applications* | | | | | | | | |
| GE | 9.8 ± 4.2 | 10.8 ± 1.8 | 2.47 ± 0.52 | 0.83 ± 0.36 | 140 ± 67 | 22.5 ± 3.3 | 32.0 ± 12.6 | 235 ± 162 |
| GE + BC | 4.8 ± 2.5 | 10.7 ± 1.6 | 2.12 ± 0.39 | 0.95 ± 0.49 | 85 ± 50 | 19.2 ± 4.4 | 21.7 ± 5.1 | 191 ± 105 |
| *5 Months After GE and GE + BC Applications* | | | | | | | | |
| GE | 17.2 ± 2.6 | 20.7 ± 6.9 | 4.93 ± 0.84 | 3.78 ± 0.38 | 95b ± 12 | 14.7 ± 6.5 | 64.7 ± 36.6 | 263 ± 75 |
| GE + BC | 18.2 ± 2.9 | 20.4 ± 2.7 | 5.70 ± 1.08 | 3.67 ± 0.51 | 100a ± 12 | 9.8 ± 6.2 | 28.8 ± 19.0 | 317 ± 99 |
| *11 Months After GE and GE + BC Applications* | | | | | | | | |
| GE | 16.2 ± 1.6 | 4.9 ± 0.6 | 2.51 ± 0.29 | 1.50 ± 0.30 | 61 ± 11 | 19.5 ± 2.4 | 83.5 ± 14.1 | 205 ± 16 |
| GE + BC | 14.3 ± 2.6 | 5.7 ± 1.2 | 2.78 ± 0.48 | 1.59 ± 0.13 | 60 ± 11 | 17.5 ± 2.4 | 88.4 ± 14.9 | 234 ± 42 |
| Control | 13.1 ± -- | 5.3 ± -- | 2.64 ± --- | 1.17 ± --- | 74 ± -- | 18.1 ± -- | 88.6 ± --- | 224 ± --- |

* Mean ± Standard Deviation; $n = 6$ for GE and GE + BC, $n = 1$ for Control; Means within a Leaf Nutrient and time not sharing the same letter are different at the 5% level.

**Table 9.** Tree heights (m) and/or DBHs (cm) at ages 5- (before treatment applications), 11- (6 months after applications), and 16- (11 months after) months of three *E. grandis* cultivars (G3, G4, G5) receiving three treatments (Green Edge only (GE), GE with biochar (GE + BC), and Control) at the IRREC biochar–fertilizer study.

| Trait: Age | Cultivar | Treatment * | | | All Treatments |
|---|---|---|---|---|---|
| | | GE | GE + BC | Control | |
| Height 5-mo | G3 | 0.8 | 1.3 | | 1.0 |
| | G4 | 1.1 | 1.3 | | 1.2 |
| | G5 | 1.0 | 1.1 | 1.0 | 1.1 |
| | All Cultivars | 1.0 | 1.2 | 1.0 | 1.0 |
| Height 11-mo | G3 | 1.6 | 2.5 | | 2.1 |
| | G4 | 2.3 | 2.7 | | 2.5 |
| | G5 | 2.2 | 2.3 | 1.4 | 2.1 |
| | All Cultivars | 2.0 | 2.5 | 1.4 | 2.2 |

**Table 9.** *Cont.*

| Trait: Age | Cultivar | Treatment * | | | All Treatments |
|---|---|---|---|---|---|
| | | GE | GE + BC | Control | |
| Height 16-mo | G3 | 2.5 | 4.8 | | 3.6 |
| | G4 | 5.1 | 5.0 | | 5.1 |
| | G5 | 4.2ab | 5.1a | 2.2b | 4.2 |
| | All Cultivars | 4.0 | 5.0 | 2.2 | 4.3 |
| DBH 16-mo | G3 | 1.5 | 4.0 | | 2.8 |
| | G4 | 4.2 | 4.2 | | 4.2 |
| | G5 | 3.5ab | 4.4a | 1.4b | 3.5 |
| | All Cultivars | 3.1 | 4.2 | 1.4 | 3.5 |

* Treatment means within Cultivar G5 not sharing the same letter differ at the 5% level.

## 4. Discussion

Eucalypts can be very productive and economically feasible when intensively grown as SRWCs, even under our preliminary assumptions. As timber markets and forestry labor are not well established in central and southern Florida, our assumed silvicultural and other forest management costs for a start-up *Eucalyptus* operation are higher than for conventional forest plantations in the South. Stumpage prices may also change as local markets develop. Deployment of elite advanced-generation *E. grandis* families may further increase profitability of SRWCs in Florida primarily due to lower seedling costs (~$0.25/seedling versus $0.70/propagule) and economic feasibility of high-yield management regimes (>2471 trees/ha).

Even under high plantation establishment and management costs, low stumpage prices, and expected coppice yields, cultivar deployment can yield positive cash flows at real discount rates greater than 10%. Under current market conditions and management costs, low-density regimes (~1181–1483 trees/ha) are the most profitable for clonal forestry on average sites (e.g., citrus lands and flatwoods). Lower propagule costs could increase financially optimum planting densities. With proper mechanical site preparation, eucalypt plantations on clay settling areas could produce higher NPVs compared to average sites [6].

The goal of our financial analysis was to demonstrate the profitability of *Eucalyptus* plantations under moderate to high discount rates and high operational costs in central and southern Florida's developing forestry markets. Since most landowners were interested in NPV and IRR, we used NPV rather than land expectation value (LEV, also known as bare land value), even though our analysis of the EH1 planting density demonstration had unequal rotation/cycle lengths. Further background on the use of LEV for *Eucalyptus* in Florida is available (6,17).

The fertilizer and planting density differences observed in the IRREC fertilizer x planting density and biochar–fertilizer studies are consistent with previously observed influences of fertilizer and planting density on eucalypt productivity in Florida [17–20] and worldwide [21–24]. While inorganic fertilizers have been necessary for rapid growth of eucalypts on Florida's infertile sandy soils, the observed response here to a slow release organic fertilizer, and its apparently beneficial coupling with BC, is encouraging for sustainable eucalypt management. Planting density effects were evident early, with, for example, the 3588 trees/ha in the fertilizer × planting density having the tallest trees at 9 months and the largest stand basal area but smallest tree DBH at subsequent ages. Similar effects of planting density have been noted for *E. dunnii* seedlings and clones [25]. Planting density trade-offs between harvest tree size, rotation length, establishment costs, and stand productivity impact plantation economics.

While our preliminary evaluation of cultivars G2 and EH1, *C. torelliana*, *E. amplifolia,* and *Q. virginiana* suggests that all appear suitable for commercial BC production in Florida, evaluations of BCs made from various woods and other feedstocks have identified that feedstock and pyrolysis condition influence properties important for using BC as a soil amendment [26,27]. GCS' new BC production

facility near Ft. Pierce, FL, scheduled to begin pilot scale testing in mid-2019, will preferably be using the cultivars G2 and EH1 and other eucalypts grown in nearby plantations. Since key objectives in BC production include minimizing the combustion of carbon, maximizing carbon content, and minimizing ash, it is imperative to ensure consistency of feedstock and the production operating environment. Known biomass characteristics, such as for the *G Series* cultivars [28], are likely to be factors in the selection of future eucalypt feedstocks.

Research on BC impact on SRWCs and forest trees in general outside Florida has generated mixed results for aspects ranging from environmental impacts to tree growth responses. When broadcast in a temperate hardwood stand in Ontario, Canada, the major short-term BC impact was an increase in limiting soil P and Ca [29]. One review of BC application in forest ecosystems found general improvements in soil physical, chemical, and microbial properties that were, however, BC-, soil-, and plant-specific [30]. A BC made from *E. marginata* decreased soil microbial carbon in a coarse soil [31], and BC added to a sandy desert soil did not significantly change soil physical properties [32]. Two BC types had different impacts on growth of young *Pinus elliottii* in subtropical China [33]. Varying doses of macadamia BC combined with two fertilizer rates had contrasting results on soil nutrients and ambiguous trends in the growth of young *E. nitens* [34]. BC did not enhance survival or growth of a *Eucalyptus* hybrid on degraded soils in southern Amazonia [35]. Compost and BC–compost mixes did not improve the performance of poplar, willow, and alder SRWCs [36]. As evidenced by presentations at a 2018 international SRWC conference [37], the biochar–fertilizer study reported here appears to be unique.

BC enhanced the soil properties of inherently poor Florida soils as well as the nutrient status of *E. grandis*, especially when applied together with organic amendments such as GE and/or chemical fertilizers. BC has a large cation exchange capacity, which facilitates retention of nutrients, particularly Ca, Mg, K, Fe, and Mn against leaching loss and thus enhances their efficient use by trees. In addition, BC has a large water holding capacity and thus improves water availability, which is especially important for Florida's sandy soils during the dry season. Due to high temperature and humidity, decomposition of organic materials in Florida's sandy soils is very rapid, and consequently these soils generally have a low organic matter content. BC can be a good organic amendment for these sandy soils, because it can stay in soil much longer than other organic materials, such as crop residues or manures.

Other potential *Eucalyptus* bioproducts may be classified as naturally occurring, generated by biochemical processes, or as the result of thermochemical processes [3,38]. Naturally occurring *Eucalyptus* bioproducts include wood products, terpenoids, phenolics, formylated phloroglucinol compounds, insecticides, repellants, antimicrobials, antifungals, and anticancers. Biorefineries such as a phosphoric lignocellulosic biorefinery [39] can produce the biochemicals lactate with parenteral and dialysis applications, succinate potentially leading to acrylic, lactic, muconic, and fumaric acids, alanine for supplements, seasonings, and antibiotics, and cellulose nanocrystals and nanofibrils for polymer nanocomposites. Sulfite paper mills and the Sulfite Pretreatment to Overcome Recalcitrant Lignin process [40] may produce jet fuel and graphene for products such as orthopedic medical implants. Thermochemical *Eucalyptus* bioproducts include biochar, syngas, and biomaterials whose carbon fiber may yield surgical implants, fabrics, filters, orthotics, chairs, beds, etc., and graphene for surgical implants, drug delivery, cancer therapy, imaging, detection of toxins, pollution, etc., graphene oxide, and batteries. These bioproducts have a broad and exciting range of applications for enhancing the value of SRWCs.

## 5. Conclusions

Two fast growing eucalypts adapted to Florida's climatic and edaphic conditions responded well to intensive culture in SRWC systems near Ft Pierce, Florida, USA. Plantations of the *E. grandis* x *E. urophylla* cultivar EH1 established on former citrus beds and managed at relatively low intensity were economically feasible; e.g., a planting density of 2071 trees/ha with three (original plus two

coppice) 5 ± year rotations resulted in an NPV in excess of $750/ha at 6% discount rate and stumpage price of $11.02/green mt. At 3,588 trees/ha, EH1 had higher stand productivity at 41 months than at lesser densities, and an organic fertilizer generally increased its growth more than an inorganic fertilizer. The organic fertilizer combined with BC increased tree sizes of three *E. grandis G Series* cultivars on an infertile sandy soil. Given that these eucalypts were determined to be suitable for the production of BC, which in turn appears to be a useful soil amendment for their intensive culture, using BC for eucalypt plantation establishment in Florida could result in more sustainable management. High-quality feedstocks such as planted eucalypts in Florida are critical to producing consistently high-quality biochar with uniform quality and specifications.

**Author Contributions:** Conceptualization, D.L.R. and M.F.E.; Methodology, D.L.R., M.F.E., K.W.F., and Z.H.; Software, D.L.R., M.F.E., K.W.F., and Z.H.; Validation, D.L.R., M.F.E., K.W.F., and Z.H.; Formal Analysis, D.L.R., M.F.E., K.W.F., R.L., F.Z., P.J., Z.Z., and Z.H.; Investigation, D.L.R., M.F.E., K.W.F., R.L., F.Z., P.J., Z.Z., and Z.H.; Resources, D.L.R., M.F.E., K.W.F., Z.H.; Data Curation, D.L.R., M.F.E., K.W.F., R.L., F.Z., P.J., Z.Z., and Z.H.; Writing—Original Draft Preparation, D.L.R., M.F.E., K.W.F., R.L., F.Z., P.J., Z.Z., and Z.H.; Writing—Review and Editing, D.L.R., M.F.E., K.W.F., R.D.C., and Z.H.; Visualization, D.L.R., M.F.E., K.W.F., Z.H.; Supervision, Z.H.; Project Administration, R.D.C.; Funding Acquisition, Z.H.

**Funding:** This research received no external funding.

**Acknowledgments:** The authors gratefully acknowledge the direct and/or indirect support provided by the IRREC, GCS, GreenTechnologies, Evans Properties, US EcoGen, Becker Tree Farm, and ArborGen.

**Conflicts of Interest:** The authors declare no conflicts of interest.

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
