# Peer review of "Short Rotation Eucalypts: Opportunities for Biochar"

_forests, doi:10.3390/f10040314_

Round 1

Reviewer 1 Report

The manuscript "Short Rotation Eucalypts: Opportunities for Bioproducts" presents data from various demonstration trials to highlight the use of Eucalyptus spp. in short rotation forestry. The organization of the results section could be improved as it is easy to confuse the two demonstrations using EH1 genotypes. My suggestion is to use to sub-headings to parse out the results from the different studies. The discussion section does not adequately put results into the context of the literature and reads more like a conclusion section. Similarly, the conclusions are not adequate to support the rest of the manuscript and should be improved. 

Specific comments:

Lines 66-67: please attempt to use the same metrics among your planting density. Either use spacing or trees per hectare

Lines 74: Using the word stage is confusing. This word choice does not need to change, but  my preference is first and second harvest   

Lines 80-81: There are formatting issues and the text exceeds the margins of the page

Lines 146-151: The font size of the text needs to be corrected.

Lines 164: E. amplifolia  and Q.virginiana do not appear in the M&M. Are these corresponding values representative of a standard or were these samples that were collected. Either way, needs more information. I'm also unfamiliar with the acronym NMF. 

 Lines 188: Were the cultivars evaluated as a composite or as individual samples? If the latter, this should reported as a mean with a standard deviation. I'd also include an "n"

Table 8 could also serve better as figure rather than a table. 

References: The reference list should be expanded to include other Eucalyptus evaluations. Alternatively, other SRWC citations. The results have not been adequately put into the context of the larger body of research on Eucalyptus and SRWC.   

Author Response

All but one of this reviewer's suggestions have been accepted as noted below:

The manuscript "Short Rotation Eucalypts: Opportunities for Bioproducts" presents data from various demonstration trials to highlight the use of Eucalyptus spp. in short rotation forestry. The organization of the results section could be improved as it is easy to confuse the two demonstrations using EH1 genotypes. My suggestion is to use to sub-headings to parse out the results from the different studies. The discussion section does not adequately put results into the context of the literature and reads more like a conclusion section. Similarly, the conclusions are not adequate to support the rest of the manuscript and should be improved. 

DONE + SUBHEADINGS ADDED

Specific comments:

Lines 66-67: please attempt to use the same metrics among your planting density. Either use spacing or trees per hectare

PLANTING DENSITY AND TREES/HA USED THROUGHOUT

Lines 74: Using the word stage is confusing. This word choice does not need to change, but  my preference is first and second harvest   

HAVE DEFINED STAGES

Lines 80-81: There are formatting issues and the text exceeds the margins of the page

DONE

Lines 146-151: The font size of the text needs to be corrected.

DONE

Lines 164: E. amplifolia  and Q.virginiana do not appear in the M&M. Are these corresponding values representative of a standard or were these samples that were collected. Either way, needs more information. I'm also unfamiliar with the acronym NMF. 

ADDED UNDER SUBHEADING 2.3

 Lines 188: Were the cultivars evaluated as a composite or as individual samples? If the latter, this should reported as a mean with a standard deviation. I'd also include an "n"

DONE

Table 8 could also serve better as figure rather than a table. 

ADDED CONTROL (0) TO TABLE 8.

References: The reference list should be expanded to include other Eucalyptus evaluations. Alternatively, other SRWC citations. The results have not been adequately put into the context of the larger body of research on Eucalyptus and SRWC.  

BOTH TYPES OF CITATIONS ADDED 

Reviewer 2 Report

The manuscript provides a valuable contribution on productivity of eucalypts in Florida as affected by planting density and soil amendments including biochar as well as the potential of these plantations as a biochar feedstock source. 

The manuscript has the makings of a good article but must be improved for clarity and completeness. The description of spacing and planting density is confusing throughout.  The Methods section lacks many needed details such as cultural treatments implemented in the EH1 demonstration planting and statistical methods utilized. The economic analysis of the cases from the demonstration planting used NPV rather than the preferred BLV even though the “planting cycles” differed with the case examined.     All tables should be reformatted omitting the unnecessary lines. Discussion should be added regarding the economic implications and the validity of the stumpage assumptions considering current and potential markets including biochar.  

See the following for specific comments.

Review Points
Forests
Short Rotation Eucalypts: Opportunities for Bioproducts
Title
Line 2-3. Change title to “Short Rotation Eucalypts: Opportunities for Biochar”. Biochar is the focus of the manuscript. Bioproducts are only mentioned in the last paragraph of the Discussion.
Abstract
Line 12-15. My understanding from the manuscript is that: 1) No cultivar was planted at 2471 trees/ha; 2) Yields were estimated for the 2471 planting density; and 3) No NPV results were reported for the 2471 trees/ha planting density. Reword abstract to better reflect results or manuscript to more clearly represent what was done.
Line 16. Produced more what?
Introduction
Line 27-28. Provide a reference on the “18 million ha in 90 countries” statement.
Line 39. E. grandis in Florida is introduced but no mention is made of the E. grandis x E. urophylla hybrid that is the subject of much of the manuscript. Mention the hybrid in the introduction.
Line 45. Delete …”which influence biochar properties” as it seems out of place.
Line 47. Specify what crops’ productivity was increased.
Material and Methods
Line 64. What cultural treatments were included in “intensively managed”?
Line 65. Specify the planting density (TPH) and corresponding spacing. The current use of planting density and spacing in the manuscript is confusing
Line 66-67. Specify planting density rather than spacing e.g. “ ..EH1 yield at 2688 trees/ha (3.1m x 1.2m) and 1536 trees/ha (3.1m x 2.1m) and estimate productivity at 2471 trees/ha”.
Line 68. Specify the size or type of inventory plot used.
Line 69. Briefly explain how yield was modeled for the two, planted densities and estimated for the 2471 trees/ha density.
Line 70. Given that planting cycle length varied with regime, land expectation value (bare land value) is more appropriate than maximum net present value.
Line 70. Explain how were yields estimated for the low and high management options.
Line 71. Specify planting density rather than spacing.
Line 74. Change to “ ….for stages 2 and 3 were assumed to be 90 and 80% of….”
Line 78. Table 1. Reorder activities sequentially.
Line 81. Spell out “Indian River Research and Education Center” at first mention in body of manuscript.
Line 81. How many trees per row?
Line 81. Change to “…receiving one of five fertilization treatments….”
Line 83. Change to “N/ha)…in combination with one of three planting densities (……..”
Line 83. Planting density (TPH) preferred to spacing.
Line 97. Change to “…in July 2017 as a randomized……”
Line 99. Change to “….within row spacing. In February…..”
Line 101. Change to “….and the interior….”
Line 122. The Methods section lacks needed information on the statistics performed on data from the fertilizer x spacing study and the windbreak study.
Results
Line 124-125. This statement is not a result. Delete or move to Introduction section.
Line 125-126. State yield values for the different planting densities.
Line 127. Delete “respectively”.
Line 128 Change to “….spacing), maximum MAI was….”
Line 130-132. Present mean DBH by density at peak MAI age.
Line 134. In Table 2, show planting densities but why do the planting densities not correspond to the stated spacing e.g. the planting density is 2688 trees/ha for a 3.1 x 1.2 m spacing. Is the 2071 trees/ha the actual planting density or the density at some time after planting. If so, the actual densities must be clarified throughout the manuscript. The trends in Max MAI and Rotation Age at Maximum MAI are logical for the current TPH reported in the table but not for the spacing and 2471 TPH reported in Table 2.
Line 134. In Table 2, change “Biological Rotation Age” to “Rotation Age at Maximum MAI”.
Line 136. Delete “Assessing the economic feasibility of EH1 SRWCs..”
Line 138. Change to “at $22.05/mt stumpage…”
Line 138. Should this be…..”Due to lower establishment and planting costs,…..”?
Line 145. In Table 3, include “years” as the unit for Stage Length column
Line 145. In Table 3, report NPV (or BLV) to the nearest dollar.
Line 145. Consider more concise alternatives to current Table 3 format.
Line 154 Include description of how to interpret letters following means.
Line 161. Text refers to hybrid as E. grandis x E. urophylla not urograndis.
Line 161. In Table 5, delete the “..of DM” in the Property column. Provide this detail in title or in footnote.
Line 165. In Table 6, provide a unit for “Recalcitrant Carbon”
Line 167-174. Were statistics done to warrant statements? Specify if or if not done in Methods. Report statistical results if done.
Line 172. Change to “….BC treatment five months….”
Line 176. In Table 7, specify units for nitrate N and ammonium N
Line 177-187. Were statistics done to warrant statements? Specify if or if not done in Methods. Report statistical results if done.
Line 201. Include description of how to interpret letters following means
Discussion
Line 204-214. Discussion is warranted on the validity of stumpage assumptions. Why did you use them? Are they reasonable?
Line 208. How does this range for “low-density regimes” fit with your results given the 3.1m x 2.1 m spacing corresponds to 1536 TPH?
Line 229. Spell out Green Carbon Solutions at first mention in body of manuscript.
Line 233. Spell out Green Carbon Solutions if starting a sentence.
Conclusions
249. Results indicated that “Low” management resulted in greater returns than “High” management but the conclusion states that eucalypts respond well to intensive culture in SRWC systems that are economically feasible. Qualify what “intensive” culture you refer to.
252. Change to “…uniform quality and specifications.”

Author Response

The manuscript provides a valuable contribution on productivity of eucalypts in Florida as affected by planting density and soil amendments including biochar as well as the potential of these plantations as a biochar feedstock source. 

VIRTUALLY ALL OF THIS REVIEWER'S SUGGESTIONS HAVE BEEN ACCEPTED AS NOTED BELOW:

The manuscript has the makings of a good article but must be improved for clarity and completeness. The description of spacing and planting density is confusing throughout.  The Methods section lacks many needed details such as cultural treatments implemented in the EH1 demonstration planting and statistical methods utilized. The economic analysis of the cases from the demonstration planting used NPV rather than the preferred BLV even though the “planting cycles” differed with the case examined.     All tables should be reformatted omitting the unnecessary lines. Discussion should be added regarding the economic implications and the validity of the stumpage assumptions considering current and potential markets including biochar.  

See the following for specific comments.

Review Points
Forests
Short Rotation Eucalypts: Opportunities for Bioproducts
Title
Line 2-3. Change title to “Short Rotation Eucalypts: Opportunities for Biochar”. Biochar is the focus of the manuscript. Bioproducts are only mentioned in the last paragraph of the Discussion.DONEAbstract
Line 12-15. My understanding from the manuscript is that: 1) No cultivar was planted at 2471 trees/ha; 2) Yields were estimated for the 2471 planting density; and 3) No NPV results were reported for the 2471 trees/ha planting density. Reword abstract to better reflect results or manuscript to more clearly represent what was done.REWORDEDLine 16. Produced more what?REWORDEDIntroduction
Line 27-28. Provide a reference on the “18 million ha in 90 countries” statement.UPDATED AND ADDEDLine 39. E. grandis in Florida is introduced but no mention is made of the E. grandis x E. urophylla hybrid that is the subject of much of the manuscript. Mention the hybrid in the introduction.DONELine 45. Delete …”which influence biochar properties” as it seems out of place.DONELine 47. Specify what crops’ productivity was increased.QUALIFIEDMaterial and Methods
Line 64. What cultural treatments were included in “intensively managed”?ADDEDLine 65. Specify the planting density (TPH) and corresponding spacing. The current use of planting density and spacing in the manuscript is confusingSTANDARDIZEDLine 66-67. Specify planting density rather than spacing e.g. “ ..EH1 yield at 2688 trees/ha (3.1m x 1.2m) and 1536 trees/ha (3.1m x 2.1m) and estimate productivity at 2471 trees/ha”.USED PLANTING DENSITY THROUGHOUTLine 68. Specify the size or type of inventory plot used.ADDEDLine 69. Briefly explain how yield was modeled for the two, planted densities and estimated for the 2471 trees/ha density.CLARIFIEDLine 70. Given that planting cycle length varied with regime, land expectation value (bare land value) is more appropriate than maximum net present value.ADDED TO DISCUSSION Line 70. Explain how were yields estimated for the low and high management options.ADDEDLine 71. Specify planting density rather than spacing.DONELine 74. Change to “ ….for stages 2 and 3 were assumed to be 90 and 80% of….”DONELine 78. Table 1. Reorder activities sequentially.DONELine 81. Spell out “Indian River Research and Education Center” at first mention in body of manuscript.DEFINED IN AUTHOR LINELine 81. How many trees per row?DONELine 81. Change to “…receiving one of five fertilization treatments….”DONELine 83. Change to “N/ha)…in combination with one of three planting densities (……..”DONELine 83. Planting density (TPH) preferred to spacing.DONELine 97. Change to “…in July 2017 as a randomized……”DONELine 99. Change to “….within row spacing. In February…..”DONELine 101. Change to “….and the interior….DONELine 122. The Methods section lacks needed information on the statistics performed on data from the fertilizer x spacing study and the windbreak study.ADDEDResults
Line 124-125. This statement is not a result. Delete or move to Introduction section.MOVEDLine 125-126. State yield values for the different planting densities.ADDRESSEDLine 127. Delete “respectively”.DONELine 128 Change to “….spacing), maximum MAI was….”DONELine 130-132. Present mean DBH by density at peak MAI age.DONELine 134. In Table 2, show planting densities but why do the planting densities not correspond to the stated spacing e.g. the planting density is 2688 trees/ha for a 3.1 x 1.2 m spacing. Is the 2071 trees/ha the actual planting density or the density at some time after planting. If so, the actual densities must be clarified throughout the manuscript. The trends in Max MAI and Rotation Age at Maximum MAI are logical for the current TPH reported in the table but not for the spacing and 2471 TPH reported in Table 2.GROSS PLOT IS WIDER THAN ROW SPACINGSLine 134. In Table 2, change “Biological Rotation Age” to “Rotation Age at Maximum MAI”.DONELine 136. Delete “Assessing the economic feasibility of EH1 SRWCs..”DONELine 138. Change to “at $22.05/mt stumpage…”DONELine 138. Should this be…..”Due to lower establishment and planting costs,…..”?RETAINEDLine 145. In Table 3, include “years” as the unit for Stage Length columnDONELine 145. In Table 3, report NPV (or BLV) to the nearest dollar.DONELine 145. Consider more concise alternatives to current Table 3 formatRETAINEDLine 154 Include description of how to interpret letters following means.DONELine 161. Text refers to hybrid as E. grandis x E. urophylla not urograndis.CHANGEDLine 161. In Table 5, delete the “..of DM” in the Property column. Provide this detail in title or in footnote.DONELine 165. In Table 6, provide a unit for “Recalcitrant Carbon”DONELine 167-174. Were statistics done to warrant statements? Specify if or if not done in Methods. Report statistical results if done.ADDEDLine 172. Change to “….BC treatment five months….”DONELine 176. In Table 7, specify units for nitrate N and ammonium NDONELine 177-187. Were statistics done to warrant statements? Specify if or if not done in Methods. Report statistical results if done.ADDEDLine 201. Include description of how to interpret letters following meansDONEDiscussion
Line 204-214. Discussion is warranted on the validity of stumpage assumptions. Why did you use them? Are they reasonable?ADDEDLine 208. How does this range for “low-density regimes” fit with your results given the 3.1m x 2.1 m spacing corresponds to 1536 TPH?CHANGEDLine 229. Spell out Green Carbon Solutions at first mention in body of manuscript.IN AUTHOR'S LINELine 233. Spell out Green Carbon Solutions if starting a sentence.USED ACRONYMN AFTER FIRST DEFINITIONConclusions
249. Results indicated that “Low” management resulted in greater returns than “High” management but the conclusion states that eucalypts respond well to intensive culture in SRWC systems that are economically feasible. Qualify what “intensive” culture you refer to.
REWORDED

252. Change to “…uniform quality and specifications.”

DONE

Reviewer 3 Report

There were a few points or questions that I would like to see cleared up.

    1.    I think that the opening sentence is really ambiguous when conversion technologies are well understood. I know most of the academia that I come in contact are certainly not aware of the different technologies of renewables. Therefore, I would suggest that the authors clarify this greatly.

    2.    The relevance of the five released cuttings through 4 generations is rather lost to the readers. Here again I suggest that the authors include the time aspect to generational intervals. At the very least I would like to see the time frame from initial testing to release of a cultivar as well as expansion time for deployment.

    3. On line 49 the sentence includes "due to their small holding capacities" The use of the term "small" should be replaced by a better descriptive term.

    4. The last sentence of the 3rd paragraph on page 2 lines 57 and 58 is certainly not verifiable thus should be omitted. It adds nothing to the paper.

    5.    On Line 33, the authors talk about damaging freezes, infertile soils, etc... However, having visited this area the term that is missing would be windthrow and that these trees seem to show very little evidence of their ability to be wind firm under he light tropical storm winds. Of course this could be side stepped by later stating that the average rotation length is less than 6 years.

    6.    The one point that kept crossing my mind is the ability to recycle nutrients and not continued fertilization. Thus, how many biochar amendments will be needed to help in this situation as well the discussion about continued fertilization with a certain portion being moved off site and causing pollution problems elsewhere.

    7.    The cost of stump of biomass seems very inflated to me even at the $11.02 figure but using the $22.05 is very far out there. Looking at Table 1, the site prep cost of $520/ac without planting is exceptionally high but your rotations are short thus driving up the ROI. The planting cost of 40 cents/tree is high as containerized pine seedlings will only cost 7 to 8 cents per tree to plant. I would like to see this explained.

    8.    The other aspect deals with the disease situation and how the selected genotypes resistance is perceived as either vertical or horizontal and the authors opinion in the effect of disease through time.

    9.    On Line 86 there is a mention of 3 other species. This needs to be clarified as there is uncertainty if the are different species of Eucs or other species?

    10.   Unless I'm missing something the difference in Biocahr between the four Eucs and live oak  compared to the PolChar was considrably differenet in pH which I suspect could be problematic   

Author Response

NEARLY ALL OF THIS REVIEWER'S SUGGESTIONS WERE ACCEPTED AS DESCRIBED BELOW:

There were a few points or questions that I would like to see cleared up.

    1.    I think that the opening sentence is really ambiguous when conversion technologies are well understood. I know most of the academia that I come in contact are certainly not aware of the different technologies of renewables. Therefore, I would suggest that the authors clarify this greatly.

REWORDED AND EXPANDED

    2.    The relevance of the five released cuttings through 4 generations is rather lost to the readers. Here again I suggest that the authors include the time aspect to generational intervals. At the very least I would like to see the time frame from initial testing to release of a cultivar as well as expansion time for deployment.

DONE

    3. On line 49 the sentence includes "due to their small holding capacities" The use of the term "small" should be replaced by a better descriptive term. 

DONE

    4. The last sentence of the 3rd paragraph on page 2 lines 57 and 58 is certainly not verifiable thus should be omitted. It adds nothing to the paper.

DONE

    5.    On Line 33, the authors talk about damaging freezes, infertile soils, etc... However, having visited this area the term that is missing would be windthrow and that these trees seem to show very little evidence of their ability to be wind firm under he light tropical storm winds. Of course this could be side stepped by later stating that the average rotation length is less than 6 years.

REWORDED

    6.    The one point that kept crossing my mind is the ability to recycle nutrients and not continued fertilization. Thus, how many biochar amendments will be needed to help in this situation as well the discussion about continued fertilization with a certain portion being moved off site and causing pollution problems elsewhere. 

NOT ADDED

    7.    The cost of stump of biomass seems very inflated to me even at the $11.02 figure but using the $22.05 is very far out there. Looking at Table 1, the site prep cost of $520/ac without planting is exceptionally high but your rotations are short thus driving up the ROI. The planting cost of 40 cents/tree is high as containerized pine seedlings will only cost 7 to 8 cents per tree to plant. I would like to see this explained. 

ADDRESSED IN DISCUSSION

    8.    The other aspect deals with the disease situation and how the selected genotypes resistance is perceived as either vertical or horizontal and the authors opinion in the effect of disease through time. 

NOT ADDRESSED

    9.    On Line 86 there is a mention of 3 other species. This needs to be clarified as there is uncertainty if the are different species of Eucs or other species?

ADDED

    10.   Unless I'm missing something the difference in Biocahr between the four Eucs and live oak  compared to the PolChar was considrably differenet in pH which I suspect could be problematic   

ADDED

Round 2

Reviewer 1 Report

General comments:

The authors have addressed many of the concerns of the previous and have greatly improved the manuscript. There remain some issues that could further improve the MS. Firstly, there is a substantial number of clunky run-on sentences that do not serve the authors or a reader. I've identified a few in specific comments. Second, the authors use DMRT to distinguish among significant differences between treatments. Because this is a post hoc test, I strongly suggest reporting the p-value or providing the results of the ANOVA either in the manuscript or in Supplementary Material. Likewise, the authors need to briefly explain why DMRT was used over a Tukey or Tukey-Kramer. DMRT is recommended for large numbers, but was used without discrimination on similarly small numbers. My third comment is a carry over from my first review. The authors have not put their results into context with the larger body of literature. How has BC affected leaf nutrients and soil properties in other studies? Have other peer-reviewed manuscripts found significant differences among heights for Eucalyptus spp grown in a variety of densities and fertilizer treatments?  This level of scholarships is needed in a discussion to understand why these results are relevant.

Specif comments:

Line 14: at 6% discount rate --> at a 6% discount rate

Line 46: water eutrophication --> eutrophication

Lines 46-50: Run-on sentence. Need to break up. 

Line 49: please change CEC to cation exchange capacity (CEC)

Lines 54-61: What knowledge gaps exist for applying biochar to SRWC in Florida and SEUSA?

Line 59-61: Why is this a separate paragraph? These last three paragraphs should be reworked as a single paragraph

Lines 68-70: Awkward run-on sentence. There seems to be a verb missing in the first half of the sentence.

Line 78: Missing a verb

Line 81-83: From where? Field trials?

Line 96: This sentence reads a little rough. Statistical analyses are completed by using SAS. It doesn't quite match to say SAS does the analysis because that still requires a person to interpret the results. 

Line 96: Why DMRT over other post hoc analyses? At what level of significance? 

Lines 107 - 111: This paragraph belongs in the next section

Line 123: Word choice is awkward. Trees aren't really permanently monumented so much as designated for sampling.

Lines 141-143: I'm not sure this is the correct post hoc test for these types of numbers. Please justify using this test.

Lines 154 - 156: Table 2: These should really be ordered from highest density to lowest at a minimum. Otherwise they seem randomly sequenced in a table.

Line 178: Just a preference, but this would be better served in the figure title

Line 207 - 208: Just a preference, but this would be better served in the figure title

Line 226: 1 for 0 --> n=1 for 0...also 0 is probably not the best shorthand here

Lines 243 - 247: Shouldn't this be in the materials and methods? It feels like I am learning this fact after I've gone through everything...

See general comments about the need for better discussion. 

Author Response

The reviewer's helpful suggestions were addressed as given below in CAPS.

General comments:

The authors have addressed many of the concerns of the previous and have greatly improved the manuscript. There remain some issues that could further improve the MS. Firstly, there is a substantial number of clunky run-on sentences that do not serve the authors or a reader. I've identified a few in specific comments. Second, the authors use DMRT to distinguish among significant differences between treatments. Because this is a post hoc test, I strongly suggest reporting the p-value or providing the results of the ANOVA either in the manuscript or in Supplementary Material. Likewise, the authors need to briefly explain why DMRT was used over a Tukey or Tukey-Kramer. DMRT is recommended for large numbers, but was used without discrimination on similarly small numbers. My third comment is a carry over from my first review. The authors have not put their results into context with the larger body of literature. How has BC affected leaf nutrients and soil properties in other studies? Have other peer-reviewed manuscripts found significant differences among heights for Eucalyptus spp grown in a variety of densities and fertilizer treatments?  This level of scholarships is needed in a discussion to understand why these results are relevant.

ALL THESE POINTS WERE CAREFULLY CONSIDERED.  SPECIFICALLY, THE MEANS FOR TABLES 7-9 WERE COMPARED USING TUKEY-KRAMER.  THE DISCUSSION WAS EXPANDED, INCLUDING 13 NEW REFERENCES.

Specif comments:

Line 14: at 6% discount rate --> at a 6% discount rate

DONE

Line 46: water eutrophication --> eutrophication

DONE

Lines 46-50: Run-on sentence. Need to break up. 

DONE

Line 49: please change CEC to cation exchange capacity (CEC)

DONE

Lines 54-61: What knowledge gaps exist for applying biochar to SRWC in Florida and SEUSA?

ADDRESSED IN DISCUSSION

Line 59-61: Why is this a separate paragraph? These last three paragraphs should be reworked as a single paragraph

COMBINED AND REORDERED 1ST TWO PARAGRAPHS.  KEPT 3RD PARAGRAPH SEPARATE TO TRANSITION BETTER TO MATERIALS AND METHODS.

Lines 68-70: Awkward run-on sentence. There seems to be a verb missing in the first half of the sentence.

REVISED

Line 78: Missing a verb

VERB IS "ASSUMED"

Line 81-83: From where? Field trials?

ADDED THIS DETAIL

Line 96: This sentence reads a little rough. Statistical analyses are completed by using SAS. It doesn't quite match to say SAS does the analysis because that still requires a person to interpret the results. 

REVISED

Line 96: Why DMRT over other post hoc analyses? At what level of significance? 

USED TUKEY-KRAMER FOR UNEQUAL n

Lines 107 - 111: This paragraph belongs in the next section

REVISED TO RETAIN POLCHAR DETAILS UNDER 2.3

Line 123: Word choice is awkward. Trees aren't really permanently monumented so much as designated for sampling.

REVISED

Lines 141-143: I'm not sure this is the correct post hoc test for these types of numbers. Please justify using this test.

USED TUKEY-KRAMER

Lines 154 - 156: Table 2: These should really be ordered from highest density to lowest at a minimum. Otherwise they seem randomly sequenced in a table.

DONE

Line 178: Just a preference, but this would be better served in the figure title

AUTHORS FEEL THAT BELOW TABLE IS LESS CLUTTERED

Line 207 - 208: Just a preference, but this would be better served in the figure title

SAME HERE

Line 226: 1 for 0 --> n=1 for 0...also 0 is probably not the best shorthand here

CHANGE "0" TO CONTROL THROUGHOUT

Lines 243 - 247: Shouldn't this be in the materials and methods? It feels like I am learning this fact after I've gone through everything...

SOME MOVED TO END OF 1ST PARAGRAPH OF 2.1

 See general comments about the need for better discussion. 

ADDED REFERENCE-RICH PARAGRAPH ON BICHAR + REFERENCES ELSEWHERE IN DISCUSSION